# Detector-in-the-Loop Tracking: Active Memory Rectification for Stable Glottic Opening Localization

**Huayu Wang**[1]                                                    HUAYU@UW.EDU
**Bahaa Alattar**[1]                                                 BALATTAR@UW.EDU
**Cheng-Yen Yang**[1]                                                CYCYANG@UW.EDU
**Hsiang-Wei Huang**[1]                                              HWHUANG@UW.EDU
**Jung Heon Kim**[2]                                                 MEDJH@DAUM.NET
**Linda Shapiro**[1]                                      SHAPIRO@CS.WASHINGTON.EDU
**Nathan White**[1,3,4]                                              WHITEN4@UW.EDU
**Jenq-Neng Hwang**[1]                                               HWANG@UW.EDU

[1] *University of Washington, Seattle, WA*

[2] *Ajou University School of Medicine, Suwon, Republic of Korea*

[3] *Harborview Medical Center, Seattle, WA*

[4] *Airlift Northwest, Seattle, WA*

**Editors:** Accepted for publication at MIDL 2026

## Abstract

Temporal stability in glottic opening localization remains challenging due to the complementary weaknesses of single-frame detectors and foundation-model trackers: the former lacks temporal context, while the latter suffers from memory drift. Specifically, in video laryngoscopy, rapid tissue deformation, occlusions, and visual ambiguities in emergency settings require a robust, temporally aware solution that can prevent progressive tracking errors. We propose Closed-Loop Memory Correction (CL-MC), a detector-in-the-loop framework that supervises Segment Anything Model 2(SAM2) through confidence-aligned state decisions and active memory rectification. High-confidence detections trigger semantic resets that overwrite corrupted tracker memory, effectively mitigating drift accumulation with a training-free foundation tracker in complex endoscopic scenes. On emergency intubation videos, CL-MC achieves state-of-the-art performance, significantly reducing drift and missing rate compared with the SAM2 variants and open loop based methods. Our results establish memory correction as a crucial component for reliable clinical video tracking. Our code will be available in `https://github.com/huayuww/CL-MR`.

**Keywords:** Video Object Detection, Video Laryngoscopy, YOLO, SAM2

## 1. Introduction

Video Laryngoscopy (VL) is a preferred method for endotracheal intubation to maintain airway patency or to stabilize oxygenation or ventilation during critical illness (Prekker et al., 2023). Automated localization and tracking of glottic opening, is a critical prerequisite for downstream tasks such as glottal area segmentation and instrument size selection(Carlson et al., 2016; Masumori et al., 2024; Matava et al., 2020; Cui et al., 2025). A popular approach in this domain is to train a single-frame detector (Wang et al., 2024; Tian et al., 2025) and apply it frame-by-frame during inference. One particular real-time model on video

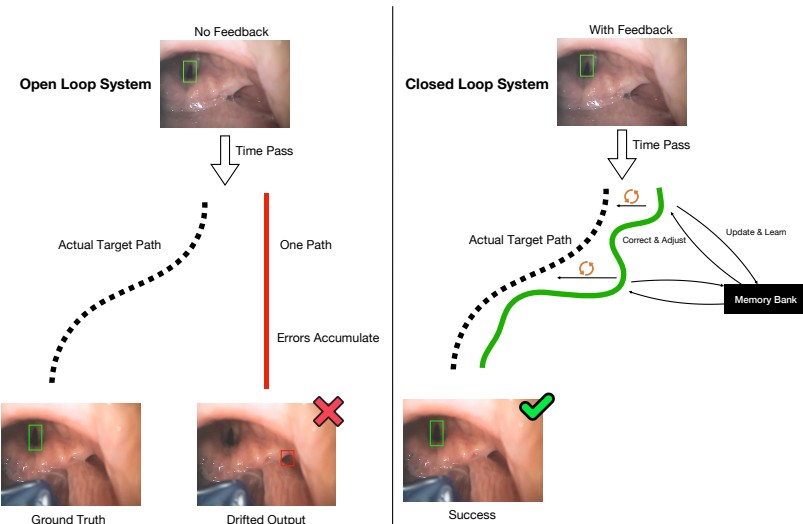

Figure 1: Comparision between open loop and closed loop tracking

laryngoscopy such as YOLO-based detectors (Kim et al., 2023) provide strong semantic discriminability and can reliably identify vocal cords from learned appearance cues. However, purely frame-wise detection is inherently unstable because it lacks temporal context, often resulting in jitter and false negatives under brief occlusions.

Traditional hybrid tracking strategies—including Kalman Filter–based association (Bewley et al., 2016; Wojke et al., 2017) and IoU-driven smoothing (Bochinski et al., 2018) purely at the output level by linking or refining detector predictions across frames. Consequently, when the tracker drifts toward semantically incorrect structures, output-level smoothing can suppress visible jitter but cannot restore the underlying memory state that caused the drift. On the other hand, video object segmentation foundation models like SAM2 (Ravi et al., 2025) deliver strong temporal continuity by using memory banks to track objects across frames (Tan et al., 2026). Yet SAM2 is class-agnostic and highly dependent on its initial prompt; in long endoscopic sequences, it is prone to semantic drift, where the tracker gradually shifts to distracting structures and cannot self-correct without external guidance as illustrated in Figure 1.

To overcome this limitation, we introduce a **Closed-Loop Memory Correction (CL-MC)** framework. Instead of treating detection and tracking as isolated components, CL-MC establishes a bidirectional pathway between a single-frame semantic detector and the SAM2 tracker, which transforms the detector's role from passive refinement to active semantic supervision, enabling drift correction without fine-tuning the foundation model and stable glottic localization under challenging clinical conditions. In summary, our contributions include **a closed-loop memory rectification mechanism** that leverages fused high-confidence detections to dynamically re-initialize SAM2's memory, providing a pathway to substantially enhance long-term video stability without fine-tuning the foundation model; **a heterogeneous confidence alignment module** that normalizes single-frame detector prediction and SAM2 prediction, enabling a unified confidence space for adaptive decision-making ; and **a state-machine–driven control strategy** designed for the challenges of endoscopic video—including occlusion, rapid deformation, and specular noise—that dynam-

ically selects the appropriate prediction source and triggers memory rectification when drift is detected.

## 2. Related Work

### 2.1. Glottic Localization in Video Laryngoscopy

Accurate localization of the glottic opening (Pedersen et al., 2023; Kruse et al., 2023) is a prerequisite for autonomous endotracheal intubation. While deep learning-based detectors like YOLO have established a strong baseline, their deployment in emergency settings is hindered by a significant data-application gap. Most public benchmarks (e.g., the Laryngoscope8 (Yin et al., 2021) dataset) are derived from transnasal laryngoscopy for diagnostic purposes. These images differ markedly from emergency intubation scenarios in terms of anatomical morphology, viewing angles, and illumination conditions. This substantial *domain shift* renders standard single-frame detectors fragile; without temporal context, they struggle to generalize to the dynamic, visually degraded environment of emergency intubation, leading to inconsistent localization and tracking instability.

### 2.2. Segment Anything Model 2 and Memory Contamination

SAM2 (Ravi et al., 2025) extends promptable segmentation to video via a sophisticated memory attention mechanism. While effective for generic tracking, SAM2 is fundamentally class-agnostic and lacks intrinsic semantic understanding of anatomical targets. In endoscopic scenarios, this reliance on low-level visual coherence makes it susceptible to *semantic drift*. Critically, SAM2 maintains temporal consistency using a First-In-First-Out (FIFO) memory bank. This passive update strategy creates a vulnerability to memory contamination: erroneous features generated during moments of blur or occlusion are indiscriminately stored and retrieved, iteratively degrading tracking performance. Although recent variants like SAMURAI (Yang et al., 2026) and MA-SAM2 (Yin et al., 2025) propose refined update rules, they remain self-contained systems without external semantic grounding. Once the memory is corrupted, these models possess no mechanism to recover, motivating our proposed active memory rectification strategy driven by high-confidence detection priors.

### 2.3. Tracking-by-Detection and Representation-Level Correction

State-of-the-art tracking frameworks such as ByteTrack (Zhang et al., 2022) and BoT-SORT (Aharon et al., 2022) associate detector outputs across frames using motion constraints or IoU-based heuristics. These methods are highly effective for multi-object tracking in natural scenes, where appearance cues are relatively stable and motion is approximately linear. However, they operate exclusively at the *bounding-box level*: detections are linked or smoothed, but the underlying representation of the tracked object remains unchanged. As a result, when the tracker deviates from the anatomical target due to occlusion, specular highlights, or rapid tissue deformation, output-level association cannot correct the internal features that caused the drift. This makes recovery particularly difficult in endoscopic video, where appearance can change abruptly and visual ambiguities are common.

In contrast, our Cl-MC method introduces a representation-level correction mechanism. High-confidence detections are not only used for frame-wise prediction but also serve as

*semantic supervisory signals* that directly update the tracker's memory. This allows the system to actively overwrite contaminated features and restore the tracker to the correct anatomical structure.

## 3. Methods

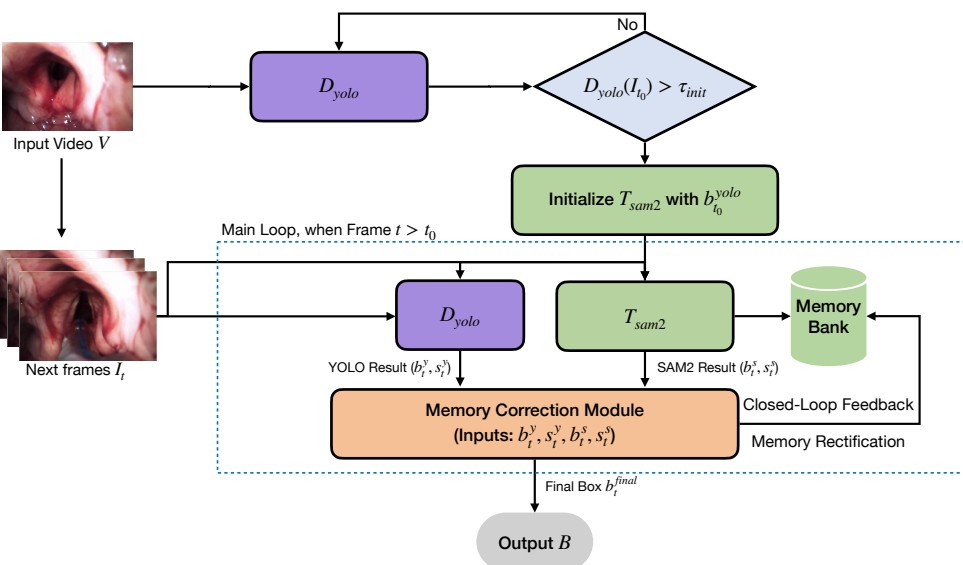

Figure 2: Upon high-confidence initialization ($\tau_{init}$), both branches process incoming frames $I_t$ to generate candidate bounding boxes and scores. The core component is the Memory Correction Module, which not only integrates predictions but also drives a Memory Rectification loop. Unlike passive FIFO updates, this state actively utilizes high-confidence fusion results to reset or refresh the SAM2 Memory Bank, thereby preventing semantic drift and memory contamination in long sequences.

We aim to extend a pre-trained single-frame glottic opening detector to the video domain without retraining. To achieve robust temporal performance under severe domain shift and endoscopic artifacts, we formulate a **Closed-Loop Memory Correction (CL-MC)** framework that integrates semantic detection with SAM2's temporal propagation. Instead of fusing predictions at the output level, CL-MC establishes an explicit control mechanism that governs when to rely on the detector, when to rely on the tracker, and when to intervene in the tracker's internal memory. The architecture consists of three key components: **(1)** a single-frame semantic detector $\mathcal{D}_{yolo}$ that provides high-confidence appearance cues; **(2)** a SAM2-based temporal tracker $\mathcal{T}_{sam2}$ responsible for visual continuity; and **(3)** a state-machine controller that aligns heterogeneous confidence signals, selects the appropriate prediction source, and triggers *memory rectification* when drift is detected as shown in Figure 2. This closed-loop design enables the system to actively overwrite corrupted memory representations, allowing stable and drift-resistant tracking of the glottic opening. The complete inference procedure is summarized in Algorithm 1.

### 3.1. Heterogeneous Confidence Alignment

A key challenge in combining predictions from the single-frame detector and SAM2 is the mismatch in their confidence semantics. The detector outputs a confidence score $s_t^y \in [0,1]$, whereas SAM2 produces a predicted IoU score $s_t^s$ reflecting mask quality. Because these signals differ in distribution and dynamic range, direct comparison is unreliable.

To obtain a unified confidence space, we apply a **Trend-aware normalization** strategy. Let $\mathcal{H}_t$ denote a sliding window containing the past $K$ tracker scores. The normalized confidence is defined as:

$$s_t^{s'} = \frac{s_t^s - \min(\mathcal{H}_t)}{\max(\mathcal{H}_t) - \min(\mathcal{H}_t) + \epsilon}. \tag{1}$$

This adaptive scaling calibrates tracker's values to local quality variations, allowing the state machine to compare $s_t^y$ and $s_t^{s'}$ on a consistent scale. The spatial consistency between predictions is evaluated using the IoU, where $v_t^{iou} = \mathrm{IoU}(b_t^y, b_t^s)$.

### 3.2. State-Machine Driven Prediction Selection

To ensure stable tracking despite occlusions, motion blur, and rapid deformation, we employ a state-machine controller that selects the appropriate source of prediction at each frame. States are determined using the aligned confidence values $(s_t^y, s_t^{s'})$ and the spatial similarity $v_t^{iou}$.

- **State 1 — Agreement:** When both predictions spatially agree

$$v_t^{iou} > \tau_{iou}, \tag{2}$$

  we refine the output using confidence-based interpolation:

$$\alpha_t = \frac{s_t^y}{s_t^y + s_t^{s'}}, \qquad b_t^{final} = \alpha_t b_t^y + (1 - \alpha_t) b_t^s. \tag{3}$$

- **State 2 — Detector Uncertain (YOLO Lost):** If the detector confidence falls below threshold

$$s_t^y < \tau_{lost}, \tag{4}$$

  the system relies on SAM2's temporal continuity:

$$b_t^{final} = b_t^s. \tag{5}$$

- **State 3 — Drift Detected (Detector Wins):** Drift is identified when the detector is confident but disagrees strongly with SAM2:

$$s_t^y > \tau_{drift} \quad \wedge \quad v_t^{iou} < \tau_{iou}. \tag{6}$$

  In this case,

$$b_t^{final} = b_t^y, \tag{7}$$

  and a memory rectification operation is triggered to correct SAM2's representation.

- **State 4 — Tracker-Preserved Conflict:** If the detector is unstable but SAM2 exhibits consistent temporal behavior,

$$b_t^{final} = b_t^s. \tag{8}$$

Rather than merging detections and tracking outputs, this controller determines *when* and *how* the tracker's internal memory should be corrected, forming the basis of our closed-loop paradigm.

### 3.3. Active Memory Rectification

Most tracking-by-detection pipelines treat the tracker as a fixed black box whose internal representation cannot be altered. In contrast, we introduce **Active Memory Rectification**, which directly intervenes in SAM2's memory bank.

Let SAM2 maintain a memory set $\mathcal{M}_t = \{m_1, \ldots, m_L\}$ at time $t$. When drift is detected (State 3), we execute a *hard reset*:

$$\mathcal{M}_t \leftarrow \text{Encode}(I_t, b_t^{final}), \tag{9}$$

thereby overwriting corrupted features with detector-guided semantics. For stable frames (States 1, 2, and 4), we apply a *soft update*:

$$\mathcal{M}_t \leftarrow \text{Update}(\mathcal{M}_{t-1}, \text{Encode}(I_t, b_t^{final})). \tag{10}$$

Importantly, the selected bounding box does not merely serve as the model output—it becomes an explicit supervisory signal used to correct or refine SAM2's representation. This closed-loop feedback mechanism enables SAM2 to recover from drift, a capability absent in conventional tracking pipelines.

## 4. Experiments

### 4.1. Dataset and Evaluation Protocol

Our semantic detector $\mathcal{D}_{yolo}$ was developed using non-emergency laryngeal images, comprising the Laryngoscope8 dataset (Yin et al., 2021) (N=2,497) and 583 clinician-annotated images curated from YouTube. For video evaluation, we utilized 24 emergency intubation sequences collected from Harborview Medical Center during prehospital air medical transports. As shown in Figure 3, the video dataset contains 8,931 frames (297 seconds) with frame-level annotations provided by an experienced clinician.

We validate our method on densely annotated laryngoscopic videos to assess robustness against aggressive corruptions, including occlusion, motion blur, and specular reflections. Following standard video detection protocols, we report mAP50, mAP50 : 95, PR-AUC(AUC), and Miss Rate, referring to the proportion of instances where the IoU with ground truth is less than 0.5.

---

**Algorithm 1** Closed-Loop Memory Correction with State-Machine Prediction Selection

---

**Input:** Video frames $\mathcal{V} = \{I_1, \ldots, I_T\}$; Detector $\mathcal{D}_{yolo}$; Tracker $\mathcal{T}_{sam}$; $\tau_{init}$; $\tau_{lost}$; $\tau_{drift}$; $\tau_{iou}$

**Output:** Target trajectory $\mathcal{B} = \{b_1, \ldots, b_T\}$

**Initialization:** Find first frame $t_0$ where $\mathcal{D}_{yolo}(I_{t_0}) > \tau_{init}$ Initialize $\mathcal{T}_{sam}$ with $b_{t_0}^{yolo}$ Initialize score history buffer $\mathcal{H}$ for normalization

**for** $t = t_0 + 1$ **to** $T$ **do**
  ▷ **Step 1: Inferencing**
  $b_t^y, s_t^y \leftarrow \mathcal{D}_{yolo}(I_t)$                                      ▷ *Detector prediction*
  $b_t^s, s_t^s \leftarrow \mathcal{T}_{sam}(I_t)$                                        ▷ *Tracker prediction*
  ▷ **Step 2: Confidence Alignment**
  $s_t^{s'} \leftarrow \text{Normalize}(s_t^s, \mathcal{H})$                          ▷ *History-aware score alignment*
  $v_{iou} \leftarrow \text{IoU}(b_t^y, b_t^s)$
  ▷ **Step 3: State-Machine Decision**
  **if** $v_{iou} > \tau_{iou}$ **then**
    |  $b_t^{final} \leftarrow \alpha \cdot b_t^y + (1 - \alpha) \cdot b_t^s$                 ▷ *State 1: Agreement*
  **else if** $s_t^y < \tau_{lost}$ **then**
    |  $b_t^{final} \leftarrow b_t^s$                             ▷ *State 2: Detector Uncertain*
  **else if** $s_t^y > \tau_{drift}$ **and** $v_{iou} < \tau_{iou}$ **then**
    |  $b_t^{final} \leftarrow b_t^y,$                        ▷ *State 3: Drift Detected* (Reset Memory)
  **else**
    |  $b_t^{final} \leftarrow b_t^s$                            ▷ *State 4: Tracker-Preserved*
  **end**
  ▷ **Step 4: Closed-Loop Feedback**
  $\text{UpdateMemory}(\mathcal{T}_{sam}, I_t, b_t^{final})$               ▷ *Inject into SAM2 memory*
**end**
**return** $\mathcal{B}$

---

### 4.2. Implementation Details

All experiments utilize official SAM2.1-Large weights. Unless stated otherwise, hyperparameters are set as follows: detector initialization threshold $\tau_{init} = 0.75$, agreement IoU $\tau_{iou} = 0.5$, drift-confidence threshold $\tau_{drift} = 0.5$. We pre-trained the YOLO12-m model $D_{yolo}$ by using the single frame dataset which mentioned in previous section. The model was initialized with COCO (Lin et al., 2014) pre-trained weights and fine-tuned for 200 epochs.

### 4.3. Baseline Models

We benchmark against two distinct tracking paradigms: (1) **Kinematic Filtering:** This category includes multiple SOTA object tracking methods such as BoT-SORT (Aharon et al., 2022) and ByteTrack (Zhang et al., 2022). These methods utilize detections solely for association via motion priors (e.g., Kalman filters), lacking mechanisms to rectify the underlying model representation when visual features degrade. (2) **Foundation Model Trackers:** We evaluate SAM2 (Ravi et al., 2025) and SAMURAI (Yang et al., 2026). While ensuring temporal continuity, these closed systems rely entirely on internal memory propagation without external semantic grounding, rendering them susceptible to cumulative drift in texture-homogeneous endoscopic environments. We applied the same point prompting method (center positive, exterior negative) to both SAM2 and SAMURAI (as

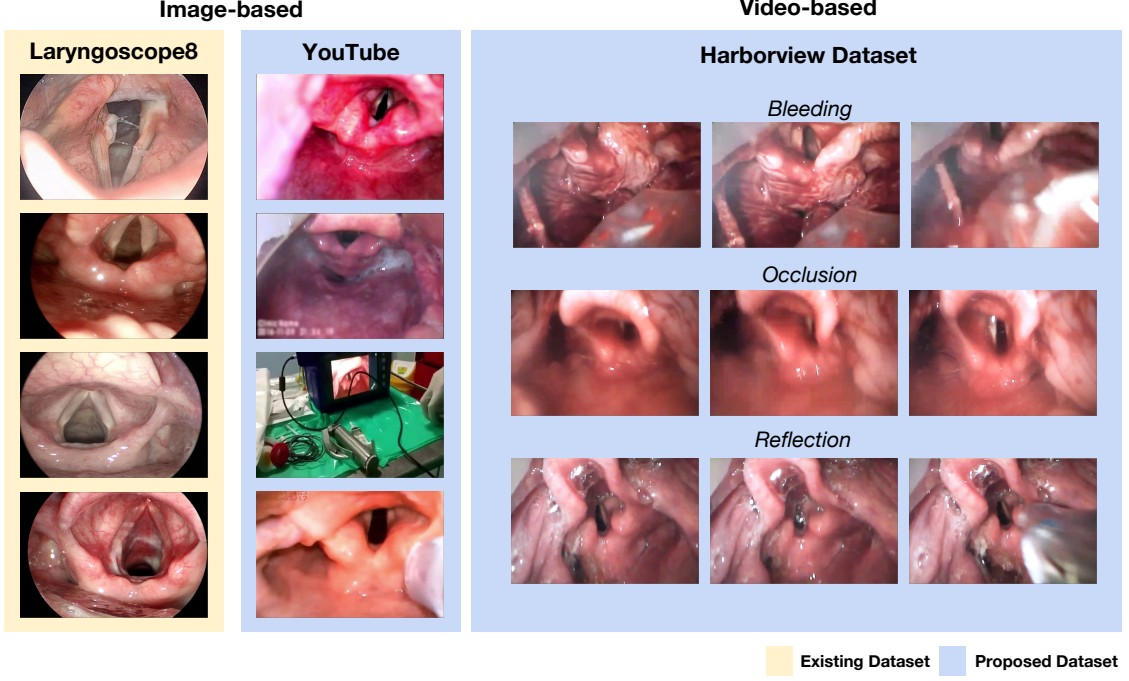

Figure 3: Detection training utilized Laryngoscope8 and YouTube image dataset, while tracking performance are evaluated on the private 26-video dataset, Harborview Dataset.

SAMURAI-Dot-Prompt), contrasting the results with SAMURAI original bounding box (as SAMURAI-Box-Prompt) configuration.

## 5. Results

### 5.1. Comparison with State-of-the-Art Methods

Table 1 shows that neither single-frame detection nor kinematic association methods provide sufficient temporal robustness for endoscopic video. SAM-based trackers improve short-term continuity but suffer from drift due to memory contamination, resulting in high missing rates. Our Closed-Loop Memory Correction (CL-MC) achieves the highest AUC and lowest missing rate by actively intervening in SAM2's memory state, demonstrating the importance of representation-level correction for reliable long-sequence tracking. Our model surpasses competing methods in $mAP_{50}$, AUC, and Missing Rate. Although we observe a marginal drop in bounding box tightness, our approach prioritizes these metrics as they offer greater clinical value for auxiliary intubation, ensuring reliable object discovery and tracking stability over strict geometric precision.

### 5.2. Ablation Studies

We conduct a component-wise ablation study to evaluate the contribution of confidence normalization and memory rectification. The tested variants are summarized in Table 2: (1) *Open-Loop Update*, which simply overwrites SAM2's memory whenever YOLO is confident;

Table 1: **Quantitative Results.** Comparison on the Harborview video dataset. All methods utilize the same YOLO detector backbone. Baselines are grouped into (1) Kinematic Association methods and (2) Foundation Model-based trackers. Our proposed closed-loop memory correction achieves the lowest missing rate and highest AUC, demonstrating its robustness under challenging conditions.

| Method | mAP$_{50}$ ↑ | mAP$_{50:95}$ ↑ | AUC↑ | Missing↓ |
|---|---|---|---|---|
| *Baseline & Kinematic Association* | | | | |
| YOLO12 | 82.41% | 51.05% | 74.57% | 8.81% |
| BoT-SORT | 82.53% | 50.20% | 74.78% | 8.33% |
| ByteTrack | 82.08% | 46.97% | 73.35% | 8.72% |
| *Foundation Model Trackers* | | | | |
| SAM2 | 70.47% | 43.18% | 66.79% | 22.24% |
| SAMURAI-Dot-Prompt | 73.61% | 48.59% | 70.42% | 20.59% |
| SAMURAI-Box-Prompt | 75.73% | **52.73%** | 70.89% | 20.25% |
| *Proposed Method* | | | | |
| **Closed-Loop Memory Correction (Ours)** | **84.32%** | 50.95% | **76.52%** | **6.85%** |

Table 2: **Effect of Proposed Components.** Component-wise analysis of the proposed method. *History Norm*: sliding-window normalization of SAM2's confidence. *Rectification*: detector-guided memory correction. Open Loop Update: YOLO directly overwrites SAM2 memory when confident. Fixed Averaging: static 0.5/0.5 fusion without confidence reasoning.

| Variant | Components | | Performance | | |
|---|---|---|---|---|---|
| | *History Norm* | *Rectification* | **mAP$_{50}$** | **AUC** | **Missing ↓** |
| Open Loop Update | × | × | 82.88% | 75.46% | 8.63% |
| Fixed Averaging (0.5/0.5) | ✓ | × | 84.15% | 76.30% | 7.01% |
| Ours (w/o Norm) | × | ✓ | 83.85% | 76.30% | 7.02% |
| **Ours (Full Model)** | ✓ | ✓ | **84.32%** | **76.52%** | **6.85%** |

(2) *Fixed Weighted Fusion*, using a static 0.5–0.5 averaging without considering model confidence; and (3) *Ours w/o Norm*, which removes the proposed trend-aware normalization and uses raw SAM2 confidence.

**Effect of Memory Rectification.** The Open-Loop and Fixed Weighted Fusion variants both lack memory correction and show noticeably higher missing rates (8.63% and 7.01%). This confirms that output smoothing alone cannot prevent progressive memory drift, and that explicit representation-level correction provides clear benefits.

**Effect of Confidence Normalization.** Removing the confidence normalization (w/o Norm) results in reduced mAP and higher instability, as SAM2's raw predicted IoU fluctuates significantly across sequences. Aligning the detector and tracker confidence spaces is crucial for triggering drift correction reliably.

**Effect of Data Scale.** Table 3 investigates the robustness of our framework compared to the baseline YOLO detector under limited data settings (30%, 70%, and 100%). While both models benefit from increased training data, our method consistently outperforms the standalone detector in most metrics. This indicates that our closed-loop tracking mechanism

Table 3: **Performance Comparison.** Our method is compared against the YOLO baseline across different data scales. The improvement relative to baseline is shown in parentheses.

| Data | mAP$_{50}$ ($\uparrow$) | | mAP$_{50:95}$ ($\uparrow$) | | Miss Rate ($\downarrow$) | |
|------|------|------|------|------|------|------|
| | YOLO | **Ours** | YOLO | **Ours** | YOLO | **Ours** |
| 30% | 64.16 | **64.79** (+0.6) | 35.09 | **36.21** (+1.1) | **21.08** | 23.55 (+2.5) |
| 70% | 81.28 | **82.29** (+1.0) | 45.98 | **46.61** (+0.6) | 11.21 | **8.31** (-2.9) |
| 100% | 82.41 | **84.32** (+1.9) | 51.05 | **50.95** (-0.1) | 8.81 | **6.85** (-2.0) |

effectively compensates for detection failures, yielding higher gains than simply scaling up the detector's training data.

**Sensitivity to Hyperparameters.** Table 4 provides a comprehensive sensitivity analysis of our framework across four key hyperparameters: the temporal window size ($T_{win}$), and thresholds for drift ($\tau_{drift}$), IoU consistency ($\tau_{iou}$), and target loss ($\tau_{lost}$). Despite evaluating all experimental configurations, the performance remains remarkably stable, with an average AUC of $76.50 \pm 0.16$ and mAP50 : 95 of $50.92 \pm 0.33$. Specifically, while slight gains are observed with a larger temporal window ($Twin = 40$) or a higher drift threshold ($\tau_{drift} = 0.7$), the framework consistently maintains its effectiveness, suggesting that it can be reliably deployed in clinical settings without the need for extensive grid-search optimization.

**Quantitative Analysis of Tracking Drift.** To investigate the primary source of tracking divergence, we conducted a component-wise performance analysis specifically on frames where tracking drift was detected. We calculated the average IoU for both the YOLO detector and the SAM 2 tracker during these drift events by comparing with ground truth. The results reveal that even when the system flags a drift, the YOLO detector maintains a relatively robust performance with an average IoU of 0.5879, whereas the SAM2 component drops to 0.4592. This performance gap confirms that tracking failures are predominantly driven by the instability of the SAM2 branch in complex scenarios.

### 5.3. Qualitative results

Qualitative results in Figure 4 illustrate the robustness of our method. While the baseline (green) is precise in simple cases (A), our method (blue) excels in complex scenarios. Panel (B) demonstrates the efficacy of our Drift Detection module, which rectifies tracker expansion caused by color similarity. Furthermore, Panel (C) shows that our method maintains accurate detection even under severe motion blur and lighting changes, significantly outperforming the baseline in conditions critical for clinical deployment.

### 6. Conclusion

We presented a closed-loop memory correction framework for reliable glottic localization in video laryngoscopy. Unlike conventional tracking-by-detection or foundation-model trackers that operate solely at the output level, our approach introduces a representation-level feedback mechanism that allows high-confidence detections to actively supervise and correct the internal memory of SAM2. Through a state-machine controller, heterogeneous confidence

Table 4: **Comprehensive Hyperparameter Analysis.** We systematically evaluate four hyper-parameters $(T_{win}, \tau_{drift}, \tau_{iou}, \tau_{lost})$. The method demonstrates high stability across all experimental configurations, achieving an average $\mathbf{mAP}_{50}$ of $84.36 \pm 0.43$, $\mathbf{mAP}_{50:95}$ of $50.97 \pm 0.29$, **AUC** of $76.53 \pm 0.12$ and **Missing Rate** of $6.77 \pm 0.23$.

| Varying Parameter | Value | Metrics | | | |
|---|---|---|---|---|---|
| | | $\mathbf{mAP}_{50}$ | $\mathbf{mAP}_{50:95}$ | **AUC** | **Missing Rate** |
| *(a) Temporal Window $T_{win}$* (Fixed: $\tau_{drift} = 0.7, \tau_{iou} = 0.5, \tau_{lost} = 0.1$) | | | | | |
| | 20 | 83.81 | 50.58 | 76.71 | 6.53 |
| Window Size | 30 | 84.87 | 51.22 | 76.67 | 6.40 |
| | 40 | 85.29 | 51.43 | 76.53 | 6.55 |
| *(b) Drift Threshold $\tau_{drift}$* (Fixed: $T_{win} = 30, \tau_{iou} = 0.5, \tau_{lost} = 0.1$) | | | | | |
| | 0.4 | 83.80 | 50.63 | 76.34 | 7.01 |
| | 0.5 | 84.32 | 50.95 | 76.52 | 6.85 |
| Threshold | 0.6 | 84.31 | 50.93 | 76.50 | 6.79 |
| | 0.7 | 84.87 | 51.22 | 76.67 | 6.40 |
| *(c) IoU Threshold $\tau_{iou}$* (Fixed: $T_{win} = 30, \tau_{drift} = 0.6, \tau_{lost} = 0.1$) | | | | | |
| | 0.4 | 84.06 | 50.49 | 76.30 | 6.94 |
| Threshold | 0.5 | 84.31 | 50.93 | 76.50 | 6.79 |
| | 0.6 | 84.06 | 51.37 | 76.67 | 7.13 |
| *(d) Lost Threshold $\tau_{lost}$* (Fixed: $T_{win} = 30, \tau_{drift} = 0.5, \tau_{iou} = 0.5$) | | | | | |
| | 0.10 | 84.32 | 50.95 | 76.52 | 6.85 |
| Threshold | 0.15 | 84.36 | 50.97 | 76.51 | 6.87 |
| | 0.20 | 84.36 | 50.97 | 76.51 | 6.87 |

alignment, and targeted memory rectification, the proposed method enables stable and drift-resistant tracking under severe domain shift, rapid deformation, and occlusion—conditions under which existing methods often fail.

Comprehensive experiments on real clinical intubation videos demonstrate that our framework consistently improves temporal robustness, achieving the highest AUC and the lowest missing rate among all baselines. Ablation studies further validate the importance of dynamic confidence normalization and memory correction, highlighting the necessity of active rather than passive temporal modeling in endoscopic video analysis.

Overall, our results indicate that closed-loop semantic feedback is a powerful and generalizable strategy for controlling foundation model trackers in medical video applications. Future work will explore extending this paradigm to multi-object anatomical tracking, leveraging richer supervisory signals, and integrating language-conditioned priors to further enhance robustness across diverse clinical environments.

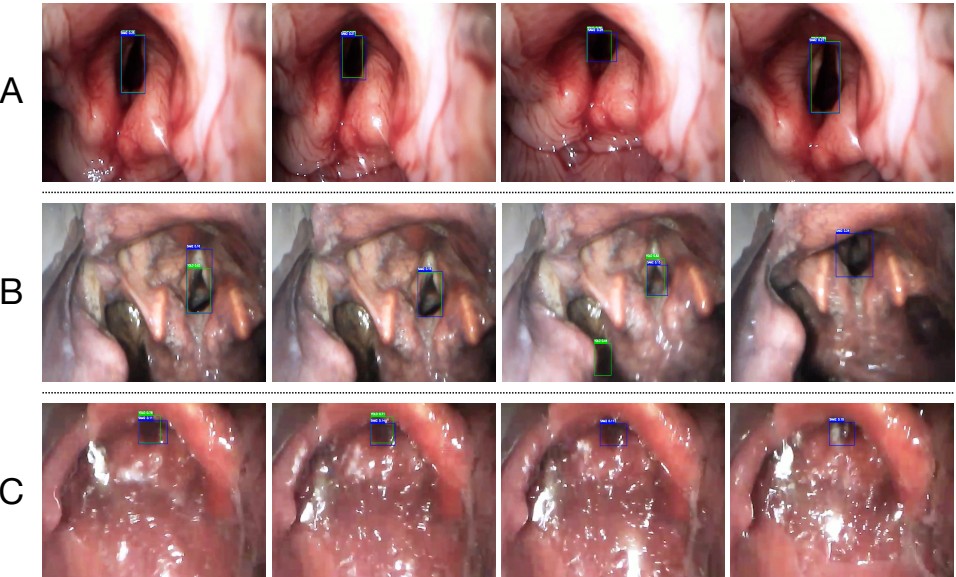

Figure 4: **Visualization of Qualitative Results.** Blue and green boxes denote the Ours and YOLO outputs, respectively.

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
