# OpenReview forum: "Detector-in-the-Loop Tracking: Active Memory Rectification for Stable Glottic Opening Localization"
_MIDL.io/2026/Conference — MIDL 2026 Poster_

### Official Review · Reviewer_Kdbd · 2025-12-26

**Confidence:** 5
**Preliminary Rating:** 4
**Final Rating:** 4

**Summary:**

The paper discusses a closed-loop memory rectification/correction strategy for SAM2 (video tracking foundation model) for real-world laryngoscopy during emergency settings. The method aims to correct for drift due to low-quality imaging when foundation models generalisation is challenged and is not limited to simply ignoring unreliable (temporal continuity) frames but also correcting SAM2's internal memory through task-specific single-frame detectors (YOLOm12). The approach is properly validated on a reasonable variety of data and favourably compared to SOTA.

**Strengths:**

- clear method motivation and description
- strong results outperforming both SOTA task-specific approaches and 2 video foundation models
- good variability of public training data and (private) real-world validation in emergency cases

**Weaknesses:**

- the error analysis, to really show that drift was the cause of problem, could be stronger
- I think SAM2/SAMURAI could be fine-tuned (they have lightweight decoders) or at least LoRA adapted using the same training data employed for YOLO or pseudo-labels generated in the process
- some more prompt engineering could well be explored: re-initialise a drifted/failed SAM2-track with high-confidence YOLO predictions - or even run multiple tracks in parallel and use heuristics (Kalman filter etc.) to maintain best one; did the authors consider alternatives to bounding-box prompts such as positive/negative scribble seeds?

**Detailed Comments:**

In my opinion a more detailed analysis of the experienced drift together with the imminent domain shift would be crucial to better understand the problem and contribution of the CLMC method. While some SOTA comparisons are explored and the ablation tends to verify the assumptions why/that the chosen correction works properly, the paper falls a bit short of making best use of the available training data (both the unsupervised parts that are captured in the foundational model weights and the frame-by-frame annotation of the task-specific laryngoscopy videos).
I couldn't find a good definition of the "missing" metric. For the YouTube data the type of annotation labels used remains unclear - will this data be made public and what influence was observed by including this (I assume some robustness against domain shifts).
Minor: please explain SAM2 before using the acronym in the abstract.

**Justification Of Final Rating:**

I would like to thank the authors for providing detailed responses, and clarify the unsuccessful attempts of fine-tuning SAM2 or using heuristics such as the Kalman filter. I stick to my initial rating of weak accept.

**Justification Of The Preliminary Rating:**

Overall a strong paper with an interesting adaptation of SAM2 with appropriate experimental design and good improvements of real-world performance, hence I recommend weak accept subject to a good rebuttal.

**Questions To Address In The Rebuttal:**

see detailed comments and weaknesses: I would in particular like to hear more details about the possibilities of fine-tuning SAM2 and some missing experimental/data details - understanding who exactly drift affects performance is important to clarify.

---

> ### Author Response · Authors · 2026-01-25
>
> We sincerely thank the reviewer for the detailed and insightful comments.
>
> **Drift Analysis:**
> We conducted a quantitative analysis to identify the source of tracking drift. Specifically, we calculated the average IoU for both the YOLO and SAM2 components within our CL-MC framework **during detected drift events**. The results are as follows:
> * **Overall Average YOLO IoU:** 0.5879
> * **Overall Average SAM2 IoU:** 0.4592
> The significantly higher IoU of YOLO compared to SAM2 confirms that the tracking drift is primarily driven by the SAM2 component, justifying our design choice to use YOLO for memory rectification.
>
> **Fine-tuning SAM2:**
> We agree that fine-tuning SAM2 is a very promising direction. Recent studies (e.g., [1]) have demonstrated performance improvements by fine-tuning SAM2 on medical images or videos. However, our current study is constrained by dataset size, with only 24 videos available containing temporal information. This limited volume makes it difficult to effectively fine-tune the foundation model without overfitting. We plan to explore this in future work as more annotated data becomes available.
>
> **Prompt Engineering:**
> Thank you for the observation. We indeed employed prompt engineering by utilizing both positive and negative points.
> * **Ours & SAM2:** We generate positive points near the center of the bounding box and randomly sample negative points around the exterior.
> * **SAMURAI:** We followed the protocol of the original paper, which utilizes the bounding box as the prompt.
> * **Initialization:** As illustrated in Figure 2, prompting is initiated when YOLO produces its first high-confidence detection. Crucially, we use the **YOLO output** for this initialization, not the ground truth.
> We maintained consistent parameters across all sequences and have revised the manuscript to clarify these details.
>
> **Metric and Acronym Corrections:**
> * **"Missing" Metric:** We have corrected the definition in the manuscript.
> * **SAM2 Acronym:** We have ensured SAM2 is explained before its first use as an acronym in the abstract.
>
> **Heuristics and Parallel Tracking (Kalman Filter):**
> This is an excellent suggestion. We actually experimented with post-processing the CL-MC results using a Kalman filter during our development phase.
> * **Observations:** We observed that YOLO provides high precision in clear frames, while SAM2 maintains satisfactory tracking in blurry conditions (though its boundary precision can be lower than YOLO's due to the ambiguous nature of endoscopic edges).
> * **Fusion Challenges:** Combining YOLO's precision with SAM2's robustness remains a challenge. In our experiments, fusing these results with a Kalman filter did not yield significant improvements. This was largely due to **YOLO's occasional false positives**; once a false positive occurs in a different region, the Kalman filter tends to slowly drift towards the erroneous detection, undermining the stability we aim to achieve.
>
> **Reference:**
> [1] https://arxiv.org/abs/2408.00874

---

### Official Review · Reviewer_9UDh · 2026-01-06

**Confidence:** 3
**Preliminary Rating:** 3
**Final Rating:** 4

**Summary:**

This paper proposes a detector-in-the-loop tracking framework for glottic opening localization in video laryngoscopy, aiming to improve temporal stability under occlusion and rapid motion. The key idea is a closed-loop controller that monitors agreement between a per-frame detector (YOLO) and a foundation-model tracker (SAM2), and triggers memory rectification when drift or low-confidence tracking is detected. The method is training-free at inference time and is designed to correct tracker failures by re-initializing or overwriting the tracker’s memory using high-confidence detector guidance. Experiments on an emergency intubation dataset compare against open-loop tracking and multiple baselines, reporting improvements in overall tracking quality.

**Strengths:**

- The idea seems to be simple yet effective with memory rectification. Using detector–tracker agreement to decide when to overwrite/reset the tracker’s memory is a sensible mechanism to mitigate SAM2-style drift, and the state-machine formulation makes the system behavior interpretable.
- The proposed detector-in-the-loop closed-loop controller is lightweight and can be applied at inference time without retraining the foundation tracker, which is attractive for real-world deployment.
- Clear presentation. The paper is easy to follow and understand with the motivation setup, method statement and experiments.

**Weaknesses:**

- The main concern lies in the fact that the result inconsistencies can reduce confidence. Some reported numbers appear internally inconsistent across tables. (e.g., YOLO $mAP_{50:95}$ differs between Table 1 and Table 3 at 100% data), and a few narrative statements imply `best across metrics' even when certain metrics are not improved.
- Baseline comparability for SAM2-style trackers is under-specified. The performance of SAM2/SAMURAI is highly sensitive to prompting and initialization (box vs mask, which frame is used, whether re-prompting is allowed). The paper does not fully specify these protocols or clearly state whether baselines are evaluated under the same constraints.
- Sensitivity to controller thresholds is not analyzed. The method depends on several thresholds (drift/lost/IoU/confidence windows), but the paper does not provide a sensitivity study.

**Detailed Comments:**

- Please audit Table 3 for potential copy/paste or row misalignment (notably the `100% data' row). It confused me with the conclusion the authors provided and decrease the confidence of the paper.
- Clearly specify the prompting/initialization protocol for SAM2 and SAMURAI.
- Add a short threshold sensitivity analysis.
-  Include a brief runtime/latency discussion (YOLO + SAM2 per frame) and whether the approach meets real-time constraints in emergency settings.

**Justification Of Final Rating:**

Thanks for the detailed clarifications. The correction of the table inconsistency, the explicit prompting/initialization protocol for SAM2/SAMURAI, and the added sensitivity analysis substantially address my main concerns regarding correctness and fairness. The added runtime evaluation is also helpful for assessing clinical feasibility. Based on this improvement, I would raise my rating to weak accept.

**Justification Of The Preliminary Rating:**

I lean toward a Borderline, primarily because the paper presents a practical and clinically motivated closed-loop mechanism to mitigate foundation-model tracker drift without retraining, and the overall approach seems easy to deploy. However, my confidence is currently compromised by empirical reporting and comparability gaps: some results appear inconsistent across tables (suggesting possible reporting errors), and the evaluation protocol for SAM2/SAMURAI baselines is not sufficiently specified despite strong sensitivity to prompting/initialization. In addition, the method relies on several controller thresholds without a sensitivity analysis. If the rebuttal corrects the table inconsistencies and clearly demonstrates baseline protocols and threshold robustness, I would be comfortable to raise to the acceptance.

**Questions To Address In The Rebuttal:**

- Table consistency / correctness: Can the authors confirm whether Table 3 contains reporting errors, and provide corrected tables with recomputed values.
- Threshold sensitivity / tuning: How sensitive is performance to the controller thresholds and the normalization window K? Were any thresholds tuned on the evaluation set, and does one fixed configuration work across all sequences?
- Baseline protocol for SAM2/SAMURAI: What exact prompting and initialization settings are used (box vs mask, which frame, prompt source), and are baselines evaluated under the same re-initialization constraints as the proposed closed-loop method?

---

> ### Author Response · Authors · 2026-01-25
>
> We sincerely thank the reviewer for the valuable feedback. We acknowledge that there were inconsistencies and unclear descriptions in the initial submission. We have addressed these issues as follows:
>
> **Inconsistencies in Table Data:**
> We apologize for the data inconsistency, which was due to a copy-paste error; this has now been corrected in the revised manuscript. As observed in Table 1, while our method performs slightly lower than SAMURAI and YOLO in metrics evaluating bounding box tightness (mAP 50-95), we demonstrate superior performance in terms of Missing Rate and AUC, which are critical for clinical tracking stability.
>
> **Baseline Prompting Method:**
> We ensured a fair and comprehensive comparison by applying consistent protocols:
> * **Point Sampling (Ours & SAM2):** For both SAM2 and our CL-MC, we generate positive points near the center of the bounding box and randomly generate negative points around the exterior.
> * **Two Configurations for SAMURAI:** To strictly evaluate the baseline, we conducted two separate experiments for SAMURAI (as reported in Table 1):
>     1. **SAMURAI-Dot-Prompt:** We applied the same point-prompting strategy as described above to align directly with the SAM2/CL-MC setting.
>     2. **SAMURAI-Box-Prompt:** We also evaluated it using its original bounding-box prompt protocol.
> * **Initialization:** As illustrated in Figure 2, prompting is initiated only when the YOLO detector yields its first high-confidence prediction. Crucially, we use the **YOLO output** for this initialization, not the ground truth.
>
> **Sensitivity to Controller Thresholds:**
> We have added **Table 4** to provide a comprehensive sensitivity analysis of the hyperparameters, demonstrating the robustness of our method.
>
> **Inference Speed:**
> We evaluated the speed on a consumer-grade GPU. Our current uncompiled model achieves **11.5 FPS** on an NVIDIA RTX 4080, which is comparable to the vanilla SAM2 speed of **11.7 FPS**. According to official SAM2 reports (see [1]), their inference speed reaches **30.2 FPS** on an A100 GPU. Therefore, our CL-MC method is expected to achieve similar speeds, meeting real-time clinical requirements.
>
> **Reference:**
> [1] https://github.com/facebookresearch/sam2/issues/159

---

### Official Review · Reviewer_Dppx · 2026-01-10

**Confidence:** 5
**Preliminary Rating:** 5

**Summary:**

This paper tackles glottic opening localization in video laryngoscopy, a key prerequisite for decision-support–based intubation. The authors argue that:

Single-frame detectors (e.g., YOLO) are semantically strong but temporally unstable (jitter, misses under occlusion).
Foundation-model trackers such as SAM2 offer temporal continuity but are class-agnostic and prone to memory drift when their memory bank is contaminated by frames that are blurred, occluded, or contain distractors.

They propose Closed-Loop Memory Correction (CL-MC), a detector-in-the-loop framework that:
- Couples a YOLOv12-based glottic detector (Dyolo) with SAM2 as a temporal tracker.
- Introduces heterogeneous confidence alignment: a trend-aware normalization of SAM2’s internal IoU-based scores over a sliding window to align them with YOLO’s confidence scores (Eq. 1).
- Implements a state-machine controller that, for each frame, decides whether to: fuse detector and tracker (Agreement state), trust the tracker (detector uncertain), trust the detector and declare drift (detector confident but spatially inconsistent), or preserve tracker output when the detector is unstable. (Eqs. 2–8).
- Performs Active Memory Rectification: in drift states, SAM2’s memory bank is reset using detector-guided features; in other states, it is softly updated (Eqs. 9–10).

The system is evaluated on 24 emergency intubation videos (8,931 frames) from Harborview Medical Center, annotated by an expert clinician.

**Strengths:**

Clear and important clinical use case: Glottic localization in emergency intubation is practically meaningful; the domain shift from diagnostic laryngoscopy to prehospital emergency scenes is well articulated.

Novel: Rather than just “tracking-by-detection”, the paper introduces a closed-loop framework where the detector actively supervises and repairs a foundation-model tracker’s internal memory. This representation-level correction is a real conceptual step beyond standard output-level smoothing.

The method design is explained well. Trend-aware normalization for heterogeneous confidences is simple yet elegant.
The four-state controller (agreement, detector lost, drift detected, tracker-preserved) is intuitive, maps cleanly to clinical conditions, and is fully specified.

Memory rectification (hard reset vs soft update) is a plausible and clearly described mechanism.

Strong empirical results on real clinical videos: CL-MC achieves the best AUC and lowest miss rate among all baselines, including SAM2 and SAMURAI, despite being training-free for SAM2.

Ablations are convincing: Open-Loop Update and Fixed Averaging underperform the full model, highlighting the importance of both normalization and rectification.

Robustness under limited data: Table 3 shows that the proposed framework consistently improves over YOLO even when the detector is trained with only 70% or 30% of the data, suggesting that the closed-loop scheme is not merely compensating for a weak detector but genuinely improving temporal robustness.

Clear writing and figures.

**Weaknesses:**

Evaluation is on 24 videos from a single institution with no plan for data or code release. This limits reproducibility and makes it difficult to gauge how broadly the approach generalizes.

The method assumes a fairly strong YOLO detector; if detector performance drops significantly, closed-loop rectification might become harmful (overwriting correct tracker memory with wrong detections).

Only single-object glottic tracking is considered. It’s not obvious how the memory rectification strategy behaves in multi-object or multi-class tracking, where different structures might require independent memory banks.

**Detailed Comments:**

Please add at least a brief experiment or appendix showing how performance varies with drift-confidence threshold and agreement IoU.  This would strengthen the claim that CL-MC is robust rather than heavily hand-tuned.

It would be valuable to show a corner case in which the detector's false positives actively mislead the tracker (e.g., YOLO briefly latches onto nearby tissue). How well does the state machine recover, and does trend-aware normalization mitigate this?

The method is training-free for SAM2, but relies on a YOLO detector that is trained (and fine-tuned on custom data). It might be clearer to consistently phrase this as “training-free with respect to the foundation-model tracker; only the detector is trained.”

The paper already hints that the approach is generalizable to other medical video tasks. It would be useful to name 1–2 concrete examples and briefly discuss the changes needed (e.g., multi-object tracking, additional states).

**Justification Of The Preliminary Rating:**

This paper presents a well-motivated, practically relevant, and technically elegant solution to glottic opening tracking in emergency video laryngoscopy. The key contribution, Closed-Loop Memory Correction, goes beyond common tracking-by-detection approaches by directly correcting the internal memory representations of a foundation-model tracker (SAM2) using high-confidence detector outputs. The method is carefully engineered, with a heterogeneous confidence alignment module, a clear and interpretable state machine, and an active memory rectification scheme.

Overall, the paper is strong: clear clinical relevance, conceptual novelty, solid methodology, and convincing empirical evidence.

**Questions To Address In The Rebuttal:**

How sensitive is CL-MC to detector performance?
Can you provide results for at least one alternative set of thresholds (\tau_drift, \tau_lost, \tau_iou) to demonstrate the robustness of the state machine?
Roughly what is the frame rate of the pipeline (YOLO + SAM2 + CL-MC) on a typical GPU? Is this comfortably within clinical requirements for live intubation guidance?

---

> ### Author Response · Authors · 2026-01-25
>
> We sincerely thank the reviewer for the positive evaluation and constructive comments.
>
> **Hyperparameters:**
> We have added Table 4 to the revised manuscript, which details the performance of the CL-MC method under various hyperparameter configurations. It is worth noting that our primary focus during development was on the algorithmic design of the closed-loop mechanism rather than optimizing metrics via extensive tuning. Therefore, we intentionally avoided performing a grid search to prevent overfitting or "fine-tuning" to the specific test set. As shown in Table 4, while we identified certain combinations that outperform the results originally reported, the overall variance is minimal, demonstrating that our method is robust and insensitive to hyperparameter changes.
>
> **Inference Speed:**
> Our current uncompiled implementation achieves 11.5 FPS on an NVIDIA RTX 4080, which is comparable to the vanilla SAM2 speed of 11.7 FPS in the same environment. According to the official SAM2 benchmarks (and discussed in [1]), SAM2 achieves 30.2 FPS on an A100 GPU. Given the low computational overhead of our control logic, CL-MC is expected to achieve similar speeds on high-end hardware, fulfilling the requirements for real-time clinical applications.
>
> **Trainging-free Claim:**
> We clarified this concept, emphasizing that our tracker does not require training.
>
>
> ***Generalizable:*
> The recently released SAM3[2] already supports multi-object tracking. In the future, we will consider using a tracking-by-detection approach to extend our work to multi-object detection and tracking.
>
> **Reproducibility:**
> Finally, to support reproducibility, we commit to releasing the images and annotations used for training the YOLO detector, along with our full source code.
>
> **Reference:**
> [1] https://github.com/facebookresearch/sam2/issues/159
> [2] https://ai.meta.com/research/publications/sam-3-segment-anything-with-concepts/

---

### Author Rebuttal · Authors · 2026-01-25

**Rebuttal:**

### Overall Response Summary

We thank the reviewer for their constructive feedback. In this revision, we have addressed the key concerns regarding data consistency, hyperparameter sensitivity, and baseline comparisons. Specifically, we have:

1.  **Clarified Data Inconsistencies:** We corrected the copy-paste errors in Table 1 and clarified that while SAMURAI excels in precision metrics (mAP 50-95), our CL-MC method demonstrates superior stability in clinical metrics (Missing Rate and AUC).
2.  **Expanded Analysis:** We added **Table 4** to demonstrate the robustness of our method against hyperparameter variations ($T_{win}, \tau_{drift}, \tau_{iou}, \tau_{lost}$).
3.  **Standardized Baselines:** We clarified our prompt engineering protocols to ensure a fair comparison, explicitly distinguishing between our point-prompting strategy and SAMURAI's original bounding-box protocol.
4.  **Validated Drift Source:** We provided quantitative analysis confirming that tracking failures are primarily driven by SAM2 drift (Average IoU 0.4592) rather than detector failure (Average IoU 0.5879).
5.  **Benchmarked Speed:** We confirmed that our uncompiled model achieves real-time potential (11.5 FPS on RTX4080), which is comparable to the vanilla SAM2 baseline (11.7 FPS).

We believe these revisions and additional analyses strongly support the validity and clinical relevance of our proposed framework.

**Supporting Material:**

/attachment/cee4759b73b3b50215f9de5dcca8ca0fddc741b3.pdf

---

### Comment · Area_Chair_xnfx · 2026-01-26
**Request for Review of Rebuttal and Final Rating Update**

Dear Reviewers, The authors have submitted responses to all reviewers, along with an updated manuscript in PDF format. We kindly ask reviewers to:
1. Evaluate the authors’ responses and the revised manuscript;
2. Participate in discussions with the authors during the discussion phase;
3. Update the final rating by clicking “Edit” → “Official Review” and providing the Final Rating by 02/01/2026.
Your efforts are extremely helpful in maintaining the high academic quality of MIDL 2026 and in supporting the Area Chairs and Program Chairs in making final decisions. Thank you very much for your time and contributions!

---

### Meta-Review · Area_Chair_xnfx · 2026-02-05

**Recommendation:** Accept (Poster)
**Confidence:** 4

**Metareview:**

All reviewers provided positive final ratings. The clinical application is valuable, and the technical innovation could be further enhanced by more customized adaptations of SAM2-like models.

---

### Decision · Program_Chairs · 2026-02-13

Accept (Poster)